# One-dimensional organic lead halide perovskites with efficient bluish white-light emission

Zhao Yuan[1], Chenkun Zhou[1], Yu Tian[2], Yu Shu[1], Joshua Messier[1], Jamie C. Wang[3], Lambertus J. van de Burgt[3], Konstantinos Kountouriotis[4], Yan Xin[5], Ethan Holt[6], Kirk Schanze[6], Ronald Clark[3], Theo Siegrist[1,2,5] & Biwu Ma[1,2,3]

Organic-inorganic hybrid metal halide perovskites, an emerging class of solution processable photoactive materials, welcome a new member with a one-dimensional structure. Herein we report the synthesis, crystal structure and photophysical properties of one-dimensional organic lead bromide perovskites, $C_4N_2H_{14}PbBr_4$, in which the edge sharing octahedral lead bromide chains $[PbBr_4{}^{2-}]_\infty$ are surrounded by the organic cations $C_4N_2H_{14}{}^{2+}$ to form the bulk assembly of core-shell quantum wires. This unique one-dimensional structure enables strong quantum confinement with the formation of self-trapped excited states that give efficient bluish white-light emissions with photoluminescence quantum efficiencies of approximately 20% for the bulk single crystals and 12% for the microscale crystals. This work verifies once again that one-dimensional systems are favourable for exciton self-trapping to produce highly efficient below-gap broadband luminescence, and opens up a new route towards superior light emitters based on bulk quantum materials.

[1] Department of Chemical and Biomedical Engineering, FAMU-FSU College of Engineering, Florida State University, Tallahassee, Florida 32310, USA. [2] Materials Science Program, Florida State University, Tallahassee, Florida 32306, USA. [3] Department of Chemistry and Biochemistry, Florida State University, Tallahassee, Florida 32306, USA. [4] Department of Physics, Florida State University, Tallahassee, Florida 32306, USA. [5] National High Magnetic Field Laboratory, Florida State University, Tallahassee, Florida 32310, USA. [6] Department of Chemistry, University of Florida, Gainesville, Florida 32611, USA. Correspondence and requests for materials should be addressed to B.M. (email: bma@fsu.edu).

Organic-inorganic hybrid metal halide perovskites, an important class of crystalline materials with excellent low-temperature solution processability and band structure tunability, have been known for decades[1–4]. They have received great attention in recent years for applications in various types of optoelectronic devices, including photovoltaic cells, light emitting diodes and optically pumped lasers[5–8]. As far as the structure is concerned, three-dimensional (3D) structure, two-dimensional (2D) layered structure, and quasi-2D structure containing layer(s) of corner sharing metal halide octahedrons have been explored extensively with much less done in one-dimensional (1D) structure, in which the metal halide octahedrons are connected in a chain[9–13].

The rich chemistry of metal halide perovskites enables numerous ways of band structure control and colour tuning by manipulating the metals, halides and organic components. Besides colour tunable narrow emissions achieved for 2D and 3D perovskites[14–19], broadband white-light emission has been realized recently in corrugated 2D perovskites with photoluminescence quantum efficiency (PLQE) of up to 9%, as a result of exciton self-trapping in the inter-layer inorganic quantum wells[20–24]. It has been well known that exciton self-trapping, due to exciton–phonon interaction, is critically dependent on the dimensionality of a crystalline system[25,26]. Lowering the dimensionality to 1D makes exciton self-trapping easier at any exciton–phonon interaction strength[27–29]. Indeed, exciton self-trapping has been observed in 1D metal halides, which exhibited broadband luminescence with large Stokes shifts of approximately 1 eV (refs 30,31). According to this principle, 1D metal halide perovskites with exciton self-trapping could be a perfect model for highly efficient white-light emitting materials, which however have not been discovered yet.

Here we report bulk and microscale crystalline 1D organic lead bromide perovskites $C_4N_2H_{14}PbBr_4$ with edge sharing octahedral lead bromide chains $[PbBr_4^{2-}]_\infty$ sitting in the columnar cages created by the $C_4N_2H_{14}^{2+}$ cations. This unique 1D structure results in strong quantum confinement with the formation of self-trapped excited states. Broadband bluish white-light emissions peaked at 475 nm with a large full width at half maximum of around 157 nm have been obtained for both the bulk and microscale crystals at room temperature, which have PLQEs of approximately 20% and 12%, respectively.

## Results

**Synthesis and characterization.** Bulk lead bromide perovskite crystals were synthesized in approximately 60% yield by reacting lead (II) bromide ($PbBr_2$) with N, N′-dimethylethylenediamine ($CH_3NHCH_2CH_2NHCH_3$) in hydrogen bromide (HBr) aqueous solution at room temperature. Microscale perovskites with the same chemical formula were also synthesized at an extremely high yield of around 95%. The structure and composition of these hybrid perovskites were fully characterized using single crystal X-ray diffraction (SCXRD), powder X-ray diffraction (PXRD), transmission electron microscopy (TEM), atomic force microscopy (AFM), proton nuclear magnetic resonance ($^1H$ NMR) and thermogravimetric analysis (TGA). Their photophysical properties were studied using steady state and time-resolved photoluminescence spectroscopies. Details of material synthesis and characterization can be found in the Methods.

**Structural characterization.** Figure 1a shows an optical microscopic image of the needle-shaped bulk 1D lead bromide perovskite crystals. Single crystalline needles as long as several centimeters can be prepared by simple solution growth. The

structure of these bulk crystals was determined using SCXRD (Supplementary Table 1). The 1D structure is depicted in Fig. 1b, with $[PbBr_4^{2-}]_\infty$ chains sitting in the rhombic columnar cages composed of $C_4N_2H_{14}^{2+}$ cations. This structure can be considered as the bulk assembly of 1D core-shell quantum wires, in which the insulating organic shells surround the core lead bromide wires (Fig. 1c). The lead bromide wires are based on double edge-shared octahedral $PbBr_6^{2-}$ units (Fig. 1d), different from the typical corner sharing octahedral $PbBr_6^{2-}$ units found in 2D and 3D metal halide perovskites[4], or mixed octahedral-trigonal prismatic $PbI_3^-$ units found in 1D hybrid lead iodides[32]. Figure 1e shows a TEM image of the microscale perovskites, with their electron diffraction pattern along [210] zone axis shown in Fig. 1f. The $d$-spacings were determined as $7.20 \pm 0.1$ Å for the (002) plane and $2.93 \pm 0.1$ Å for the (121) plane, consistent with the values obtained from SCXRD for the bulk crystals. The PXRD patterns of the microscale crystals show almost identical features as for the ball-milled powders of the bulk single crystals (Fig. 1g). All these results confirm that the microscale perovskite crystals prepared by the one-pot synthesis possess the same composition and crystal structure as the bulk single crystals prepared by solution growth. $^1H$ NMR, TGA and AFM were also used to characterize the microscale perovskite crystals and the results can be found in the Supplementary Figs 1, 2 and 3, and Supplementary Table 2. Instead of the needle shape observed in the bulk crystals, the microscale crystals display a hexagonal shape, with a lateral dimension size of 0.5–2 μm, and a thickness of 1–1.5 μm. The relatively short reaction time in one-pot synthesis is believed to suspend the extended crystallization along the metal halide chain direction.

**Photophysical properties.** We have investigated the photophysical properties of the prepared bulk and microscale 1D lead bromide perovskites. Figure 2a,b show the photo-images of the bulk perovskite crystals under ambient light and UV (365 nm) irradiation, respectively. We find that the needle-shaped crystals are colourless and show strong bluish white-light emission upon UV irradiation. The absorption and emission spectra of the bulk and microscale crystals are shown in Fig. 2c. High energy absorption with peaks at around 375 nm was observed for both the bulk and microscale crystals. Upon excitation at 360 nm, both the bulk and microscale crystals exhibited broad emissions, with a maximum at 475 nm and a large full width at half maximum of 157 nm (0.90 eV), similar to the features of the corrugated 2D lead bromide perovskite (EDBE)[PbBr$_4$] (ref. 21). Blue shift of the absorption and emission (approximately 100 nm) spectra of these 1D perovskites as compared to the corrugated 2D perovskites is due to the stronger quantum confinement in the 1D structure[33]. Figure 2d shows the decay of emissions at 475 nm from the bulk and microscale crystals at room temperature, giving lifetimes of 37.3 and 26.6 ns, respectively. In the microscale crystals, a well-defined emission peak at 389 nm was observed, which becomes much less intense in the bulk crystals. This may be attributed to the stronger self-absorption in bulk crystals. The 389 nm emission peak returns for the ball-milled single crystal samples, where the particle size is close to that of microscale crystals (Supplementary Fig. 4). The slightly red-shifted absorption in the bulk crystals as compared to the microscale crystals also suggests stronger self-absorption at the band edge. Considering the similarity of the photophysical properties between our 1D perovskites and the corrugated 2D perovskites[20–23], it is reasonable to assign the high energy narrow emission to the free excitons and the

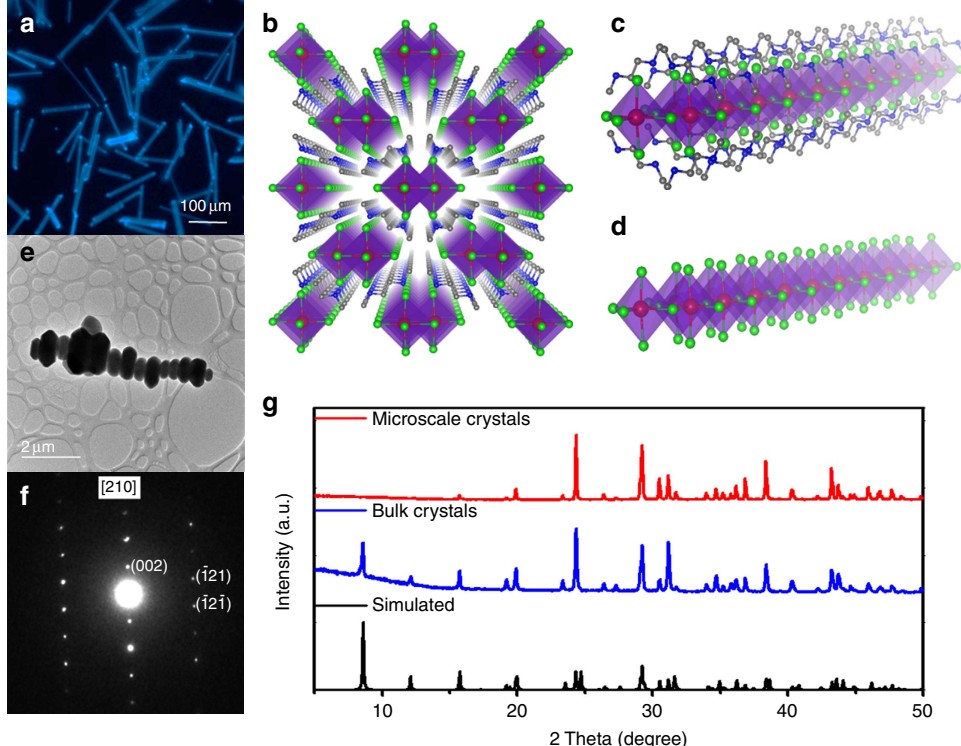

**Figure 1 | Structural characterization of bulk and microscale 1D lead bromide perovskite crystals.** (**a**) Optical image of needle-shaped 1D lead bromide perovskite single crystals. (**b**) Structure of 1D perovskite $C_4N_2H_{14}PbBr_4$ (red spheres: lead atoms; green spheres: bromine atoms; blue spheres: nitrogen atoms; grey spheres: carbon atoms; purple polyhedrons: $PbBr_6^{4-}$ octahedrons; hydrogen atoms were hidden for clarity). (**c**) View of an individual lead bromide quantum wire wrapped by the organic cations. (**d**) View of an individual lead bromide quantum wire with edge sharing octahedrons. (**e**) Transmission electron microscopy (TEM) image of the microscale 1D perovskite crystals. (**f**) Electron diffraction pattern along [210] zone axis. (**g**) Powder X-ray diffraction (PXRD) patterns of the bulk and microscale 1D perovskite crystals, as well as the simulated PXRD patterns based on the single crystal structure.

low energy broad emission to the self-trapped excitons. Identical excitation spectra for peak emissions at 389 and 475 nm (Supplementary Fig. 5), as well as the excitation independent emission spectra (Supplementary Fig. 6) suggest that both the high energy narrow emission and the low energy broad emission originate from the same excited states. The same decay curves at 389 and 475 nm for the microscale crystals further confirm that the free exciton and self-trapped excited states are genuinely in equilibrium at room temperature (Supplementary Fig. 7). This dual emission behaviour is indeed similar to what have been observed in the phosphorescent molecular butterflies, in which the excited state potential energy surface has two energy minima due to photoinduced structural changes[34,35].

The Comission Internationale de l'Eclairage (CIE) chromaticity coordinates of the overall light emission were determined as (0.21, 0.28) and (0.21, 0.27) for the bulk and microscale perovskites (Fig. 2e), which give correlated colour temperatures of 21,242 and 24,154 K, respectively, corresponding to so-called cold bluish white light. The broad emissions also lead to good colour-rendering indexes of 63 and 66 for the bulk and microscale crystals, respectively, which are comparable to the colour-rendering indexes of typical fluorescent light sources (approximately 65). The PLQEs of bulk and microscale crystals were measured to be 18–20% and 10–12%, respectively (Supplementary Fig. 8). These materials also show moderate photostability in air with slow decrease of emission intensity under continuous high power Hg lamp irradiation (Supplementary Fig. 9). Based on the PLQEs and lifetimes, the radiative decay rates were calculated to be approximately

$0.5 \times 10^7 \, s^{-1}$ and $0.4 \times 10^7 \, s^{-1}$, and the nonradiative decay rates of approximately $2.2 \times 10^7 \, s^{-1}$ and $3.3 \times 10^7 \, s^{-1}$ for the bulk and microscale crystals, respectively. The higher nonradiative decay rate for the microscale crystals is not surprising if one considers the much larger surface areas, where the metal halide quantum wires might not be perfectly wrapped by the organic cations to create self-trapped excited states. The larger surface areas could also lead to more surface defects to act as nonradiative decay channels. Table 1 summarizes the major photophysical properties for the bulk and microscale 1D lead bromide perovskite crystals.

## Discussion

To verify the origin of the broadband emission from the intrinsic self-trapped excitons, we have measured the dependence of room-temperature luminescence intensity on excitation power for the bulk and microscale crystals, as well as their emissions at different temperatures from room temperature to 77 K. As shown in Fig. 3a, the intensity of the broadband emission exhibits a linear dependence on the excitation intensity up to 500 W cm$^{-2}$ (490 nJ per laser pulse), suggesting that emission does not arise from permanent defects[21]. For the temperature dependent emissions of both the bulk and microscale crystals, a smooth progression in favour of the low energy emission band has been observed as the temperature is lowered (Supplementary Fig. 10). At 77 K, the bulk and microscale crystals exhibited broadband, Gaussian-shaped and strongly Stokes shifted luminescence (Fig. 3b), with the same features as those of self-trapped

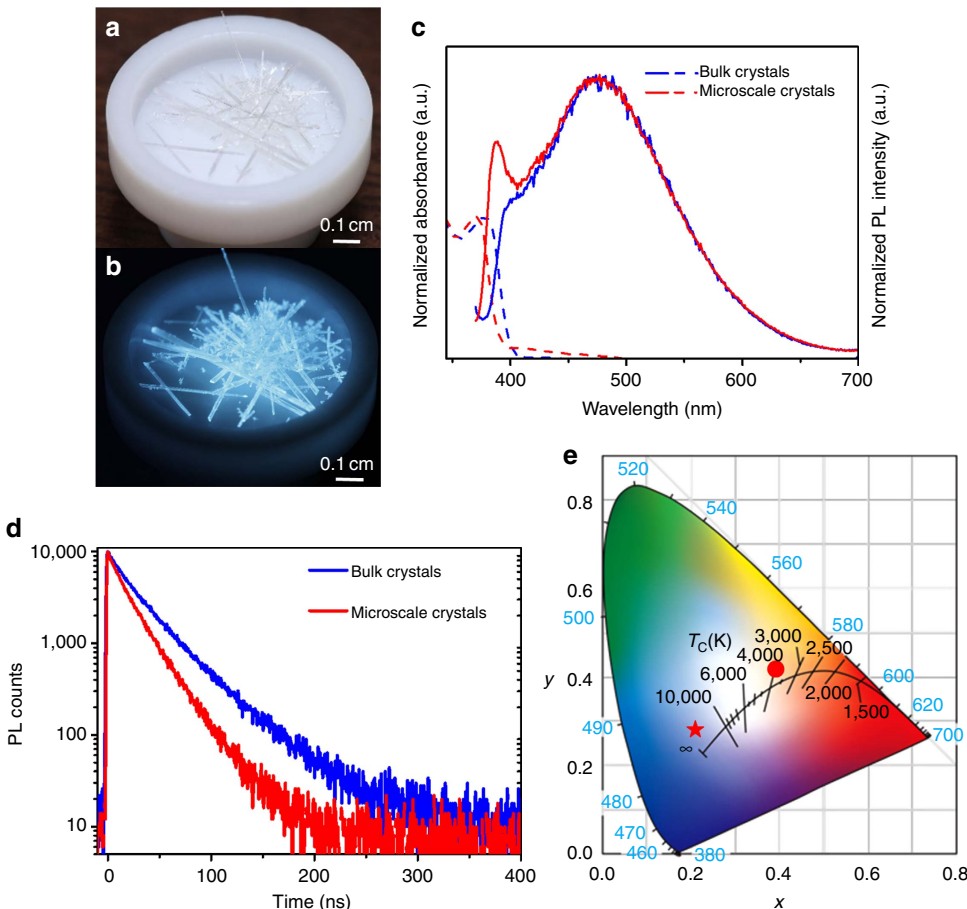

**Figure 2 | Photophysical properties of 1D lead bromide perovskites at room temperature.** (**a**) Image of bulk perovskite crystals under ambient light. (**b**) Image of bulk perovskite crystals under UV light (365 nm). (**c**) Absorption (dash lines) and emission (solid lines, excited at 360 nm) spectra of the bulk and microscale perovskite crystals at room temperature. (**d**) The photoluminescence decays of the bulk and microscale 1D perovskite crystals (measured at 475 nm) at room temperature. (**e**) Comission Internationale de l'Eclairage (CIE) chromaticity coordinates of the 1D perovskites in this work (star), and the corrugated 2D perovskite (EDBE)[PbBr$_4$] (ref. 21) (circle).

**Table 1 | Photophysical properties of the bulk and microscale 1D lead bromide perovskite crystals.**

| 1D perovskites | $T$ (K) | $\lambda_{abs}$ (nm) | $\lambda_{em}$ (nm) | $\phi$(%) | $\tau_{av}$ (ns) | $k_r$ (s$^{-1}$) $\times 10^7$ | $k_{nr}$ (s$^{-1}$) $\times 10^7$ |
|---|---|---|---|---|---|---|---|
| Bulk crystals | 295 | 379 | 475 | 18–20 | 37.3 | 0.51 | 2.17 |
|  | 77 | NA | 525 | NA | 1,443.6 | NA | NA |
| Microscale crystals | 295 | 371 | 475 | 10–12 | 26.6 | 0.41 | 3.35 |
|  | 77 | NA | 525 | NA | 1,347.4 | NA | NA |

$\lambda_{abs}$ is the wavelength at absorbance maximum; $\lambda_{em}$ is the wavelength at the emission maxima; $\phi$ is the PL quantum efficiency; $\tau_{av}$ is the PL lifetime; $k_r$ and $k_{nr}$ are the radiative and non-radiative decay rates calculated from equations, $k_r = \phi/\tau$ and $k_{nr} = (1-\phi)/\tau$.

excitons observed in many metal halide crystals at low temperature and corrugated 2D halide perovskites[20–23]. In those cases, electron-spin resonance measurements and *ab-initio* calculations have identified Pb$_2$$^{3+}$, Pb$^{3+}$ and X$_2^-$ (X = Cl, Br) as the potential radiative self-trapped states[24,25]. Figure 3c shows the emission decays at 77 K, with long lifetimes calculated to be 1,443.6 and 1,347.4 ns for the bulk and microscale crystals, respectively. Consistent with the decay rates measured at room temperature, the microscale crystals possess slightly higher nonradiative decay rate than the bulk crystals at 77 K.

The exciton self-trapping in the 1D lead bromide perovskites can therefore be described in the configuration coordinate diagram given in Fig. 3d (refs 22–24): at room temperature, the free excitons and self-trapped excitons coexist due to an equilibrium created by thermal activation. This results in the bluish white-light emission with two bands: a high energy narrow band of free excitons and a low energy broad band of self-trapped excitons. At 77 K (low thermal activation), all excitons are in self-trapped states, resulting in only strongly Stokes-shifted broadband luminescence. This model also offers a natural explanation of the observation that the emission spectra for the bulk and microscale crystals are different at room temperature, but identical at 77 K. At room temperature, the self-absorption in the bulk crystals is stronger than in the microscale crystals, leading to lower emission intensity from the free excitons at approximately 389 nm. However, at 77 K, all the excitons are in the self-trapped states for both the bulk and microscale crystals, resulting in the same emissions without self-absorption.

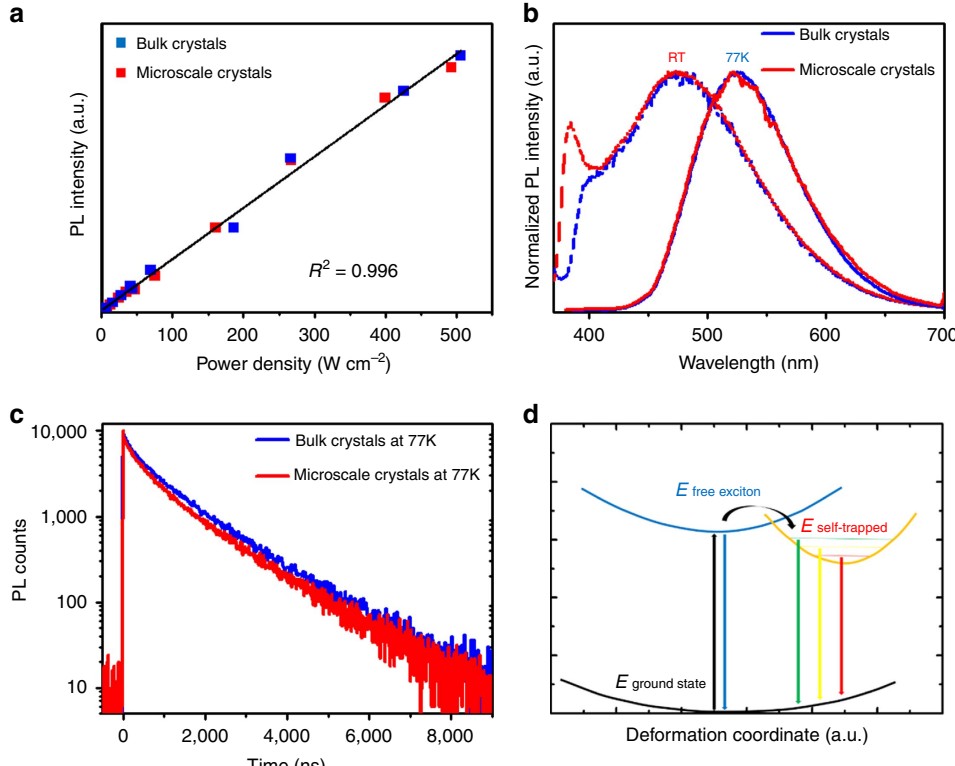

**Figure 3 | Verification of the photoluminescence mechanism of exciton self-trapping in 1D perovskites.** (**a**) PL intensity versus excitation power for both the bulk and microscale 1D perovskite crystals at room temperature. (**b**) Emissions of the bulk and microscale 1D perovskite crystals (excited at 360 nm) at room temperature (dash lines) and 77 K (solid lines). (**c**) Luminescence decays of the bulk and microscale 1D perovskite crystals at 77 K (measured at 525 nm). (**d**) Configuration coordinate diagram for the coexisting of free and self-trapped excitons in 1D perovskites; the straight and curved arrows represent optical and relaxation transitions, respectively.

By carefully choosing the organic cation, we have been able to design and synthesize single crystalline 1D lead bromide perovskites by wet chemistry methods with high yield and excellent reproducibility. Our work advances the research on organic-inorganic hybrid metal halide perovskites from well-studied 3D and 2D structures to a novel 1D structure. The highly luminescent 1D bulk quantum materials reported here, with an unusual broadband emission, are promising light emitters for optoelectronic devices, which can also serve as a platform for fundamental studies of the structure-property relationships in bulk quantum wire systems.

## Methods

**Materials.** Lead (II) bromide (99.999%), N, N'-dimethylethylenediamine (99%), hydrobromic acid (48 wt.% in $H_2O$), octanoic acid (98%) and hexane (98.5%, mixture of isomers) were purchased from Sigma-Aldrich. All reagents and solvents were used without further purification unless otherwise stated. Spectroscopic grade solvents were used in the UV-Vis and photoluminescence spectroscopic measurements.

**Growth of $C_4N_2H_{14}PbBr_4$ bulk single crystals.** Lead(II) bromide (0.100 g, 0.27 mmol) and N, N'-dimethylethylenediamine (0.024 g, 0.27 mmol) were combined in 10 ml of 48 wt. % hydrobromic acid and sonicated for 10 min to obtain a clear solution. One millilitre of the stock solution was set up in vapour diffusion chambers with acetone. Colourless needle-like crystals of $C_4N_2H_{14}PbBr_4$ were obtained through diffusion of acetone into this solution over a 24 h period. The crystals suitable for X-ray structure determination were washed with acetone, and dried under reduced pressure to afford 0.010 g (yield of approximately 60%) of product. Element analysis: $C_4N_2H_{14}PbBr_4$, calculated C, 7.79%; H, 2.29%; N, 4.54%; found C, 7.81%; H, 2.18%; N, 4.44%.

**Synthesis of $C_4N_2H_{14}PbBr_4$ microscale 1D perovskites.** 0.1 mmol Lead(II) bromide (36.7 mg) was added in 5 ml hexane solution with 1.0 mmol octanoic acid (160 µl), followed by injecting 0.3 mmol N, N'-dimethylethylenediamine (32 µl) to form a turbid solution. The reaction solution was vigorously stirred at room temperature for 30 min until a white colloidal solution was formed and the white microscale perovskites were obtained by centrifugation to remove the unreactive materials in the clear supernatant, affording a white powder in a yield of around 95% after having dried under vacuum.

**Optical imaging.** Optical micrograph was obtained using an inverted Nikon Ti epifluorescence microscope equipped with an Andor iXonEM+ 885 EMCCD camera.

**Single crystal X-ray diffraction (SCXRD).** A needle-shaped crystal was mounted on a nylon loop with the use of heavy oil. The sample was held at −170 °C for data collection. The data were taken on a Bruker SMART APEX II diffractometer using a detector distance of 6 cm. The number of frames taken was 2,400 using 0.3 degree omega scans with either 20 or 30 s of frame collection time. Integration was performed using the program SAINT which is part of the Bruker suite of programs. Absorption corrections were made using SADABS. XPREP was used to obtain an indication of the space group and the structure was typically solved by direct methods and refined by SHELXTL. The non-hydrogen atoms were refined anisotropically. A matrix was run at 253 K (−20 °C) to check on whether a phase change could have occurred with the structure run at low temperatures. The index was essentially the same as we found earlier at low temperature, ruling out a phase change.

**Powder X-ray diffraction (PXRD).** The PXRD analysis was performed on a Panalytical X'PERT Pro Powder X-Ray Diffractometer using Copper X-ray tube (standard) radiation at a voltage of 40 kV and 40 mA, and X'Celerator RTMS detector. The diffraction pattern was scanned over the angular range of 5–50 degrees ($2\theta$) with a step size of 0.02, at room temperature. Simulated powder patterns were calculated by Mercury software using the crystallographic information file from the single-crystal X-ray experiment.

**Transmission electron microscopy (TEM) images.** Microstructural characterization was performed using TEM, on a JEOL JEM-ARM200cF at 200 kV. Low intensity illumination and fast acquisition time were used during data collection to avoid beam damage. TEM samples were prepared by depositing a few drops of the microscale perovskite solution on a carbon film supported copper grid (200 mesh); the samples were subsequently dried overnight.

**Atomic force microscopy (AFM) images.** AFM measurements were conducted using Bruker Icon. All measurements were performed in the standard tapping mode with OTESPA-R3 tips from Bruker.

**Proton nuclear magnetic resonance ($^1$H NMR).** $^1$H NMR spectra were acquired at room temperature on Bruker AVANCE III NMR Spectrometers with a 500 MHz Bruker magnet. All chemical shifts ($\delta$) were reported in ppm relative to tetramethylsilane.

**Thermogravimetry analysis (TGA).** TGA was carried out using a TA instruments Q50 TGA system. The samples were heated from room temperature (around 22 °C) to 800 °C at a rate of 5 °C min$^{-1}$, under a nitrogen flux of 100 ml min$^{-1}$.

**Absorption spectrum measurements.** Absorption spectra of both bulk and microscale perovskite crystals were measured at room temperature through synchronous scan in an integrating sphere incorporated into the spectrofluorometer (FLS980, Edinburgh Instruments) while maintaining a 1 nm interval between the excitation and emission monochromators.

**Photoluminescence steady state studies.** Steady-state photoluminescence spectra of both the bulk and microscale crystals in solid state were obtained at room temperature and 77 K (liquid nitrogen was used to cool the samples) on a Varian Cary Eclipse Fluorescence spectrophotometer.

**Photoluminescence quantum efficiencies (PLQEs).** For PLQE measurement, the samples were excited using light output from a housed 450 W Xe lamp passed through a single grating (1,800 l mm$^{-1}$, 250 nm blaze) Czerny-Turner monochromator and finally a 5 nm bandwidth slit. Emission from the sample was passed through a single grating (1,800 l mm$^{-1}$, 500 nm blaze) Czerny-Turner monochromator (5 nm bandwidth) and detected by a Peltier-cooled Hamamatsu R928 photomultiplier tube. The absolute quantum efficiencies were acquired using an integrating sphere incorporated into the FLS980 spectrofluorometer. The PLQE was calculated by the equation: $\eta_{QE} = I_S/(E_R - E_S)$, in which $I_S$ represents the luminescence emission spectrum of the sample, $E_R$ is the spectrum of the excitation light from the empty integrated sphere (without the sample) and $E_S$ is the excitation spectrum for exciting the sample. The PLQEs were double confirmed by a Hamamatsu C9920 system equipped with a xenon lamp, calibrated integrating sphere and model C10027 photonic multi-channel analyser. Control samples, Rhodamine 101 and blue phosphor BaMgAl$_{10}$O$_{17}$:Eu$^{2+}$, were measured using this method to give PLQEs of approximately 98 and 93%, which are close to the literature reported values.

**Time-resolved photoluminescence.** Time-resolved emission data were collected at room temperature and 77 K using the FLS980 spectrofluorometer. The dynamics of emission decay were monitored by using the FLS980's time-correlated single-photon counting capability (1,024 channels; 1 μs window) with data collection for 10,000 counts in the maximum channel. Excitation was provided by an Edinburgh EPL-360 ps pulsed diode laser. The average lifetime was obtained from the tri-exponential decays according to equation (1).

$$\tau_{ave} = \sum \alpha_i \tau_i^2 / \sum \alpha_i \tau_i, \ i = 1, 2, 3 \tag{1}$$

where $\tau_i$ represents the decay time and $\alpha_i$ represents the amplitude of each component.

**PL intensity dependence on excitation power density.** PL intensity versus power studies were carried out on an Edinburgh Instruments PL980-KS transient absorption spectrometer using a Continuum Nd:YAG laser (Surelite EX) pumping a Continuum Optical Parametric Oscillator (Horizon II OPO) to provide 360 nm 5 ns pulses at 1 Hz. The pump beam profile was carefully defined by using collimated laser pulses passed through an iris set to 5 mm diameter. Pulse intensity was monitored by a power meter (Ophir PE10BF-C) detecting the reflection from a beam splitter. The power meter and neutral density filters were calibrated using an identical power meter placed at the sample position. Neutral density filters and an external power attenuator were used to reduce the pump's power density to the desired power range. Detection consisted of an Andor intensified CCD (1,024 × 256 element) camera collecting a spectrum from 287 to 868 nm and gated to optimize PL collection (typically a 30-50 ns gate depending on PL lifetime starting immediately following the 5 ns laser pulse). One hundred collections were averaged at each power level with every laser pulse monitored to determine the average intensity. PL intensity was determined at the maximum of the PL emission curve. Peak power density (W cm$^{-2}$) was calculated by dividing the observed pulse intensity (in Joules) by the pulse width ($5 \times 10^{-9}$ s) and the beam area in cm$^2$ (area = $\pi r^2 = \pi(0.25 \text{ cm})^2 = 0.20 \text{ cm}^2$). For example, 500 W cm$^{-2}$ peak power density was derived from an observed 490 nJ pulse intensity.

**Temperature-dependent photoluminescence.** Measurements of photoluminescence at different temperatures were performed on both the bulk and microscale perovskite crystals dispersed and sandwiched between glass microscope slides (1 × 2.5 cm). Samples were held inside an OptistatDN variable liquid nitrogen cryostat (Oxford Instruments) at a 45° angle to the excitation source by a 2 cm × 2 cm sample holder fixed to the end of a cryostat sample holder. The emission spectra were collected at 90° to the excitation source with a Photon Technology International (PTI) spectrophotometer with photomultiplier detection system. The monochromatic excitation ($\lambda = 350$ nm) was filtered through a 350 nm bandpass filter to reduce noise from spurious wavelengths. Individual spectra were recorded at intervals of 10 K with the aid of an Omega CYC3200 autotuning temperature controller and thermocouple wire affixed to the sample glass slide. Dry nitrogen was passed through the sample compartment to avoid condensation on the outer cryostat windows.

**Photostability study.** To test the material photostability, a 100 W 20 V mercury short arc lamp was used as the continuous irradiation light source. The intensity of the irradiation light source was calibrated to 150 mW cm$^{-2}$. The photoluminescence was measured at periodic intervals on a HORIBA iHR320 spectrofluorometer, equipped with a HORIBA Synapse CCD detection system.

**Data availability.** The data that support the findings of this study are available from the corresponding author on request.

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

## Acknowledgements

We thank the Florida State University for financial support through the Energy and Materials Initiative. Part of the work was performed at the TEM facility at the National High Magnetic Field Laboratory, which is supported by the FSU Research Foundation, the National High Magnetic Field Laboratory (NSF-DMR-0654118) and the State of Florida. The excitation power dependent luminescence measurements were performed on a transient absorption spectrometer supported by the National Science Foundation under Grant No. CHE-1531629. Jamie Wang is supported by the National Science Foundation Graduate Research Fellowship under Grant No. DGE-1449440. The authors acknowledge Dr Lei Zhu for providing access to a fluorometer, Dr Jingjiao Guan and Junfei Xia for help with the optical images, Dr Hanwei Gao for the access to the instruments for photostability test, Dr Peter Djurovich for help with PLQE measurements, and Dr Peng Xiong for helpful discussions.

## Author contributions

Z.Y. and B.M. conceived the experiments and analysed and interpreted the data. Z.Y., C.Z. and J.M. synthesized and purified the bulk and microscale crystals; Z.Y. collected Powder XRD, proton NMR and TGA data; Y.S. performed AFM study; Z.Y., C.Z., J.C.W. and K.K. measured the photophysical properties; Y.T. performed the photostability test; Z.Y., C.Z. and L.J.B. performed PL intensity versus excitation power studies; E.H. and K.S. performed temperature-dependent emission measurement; R.C. performed single crystal XRD analysis; Y.X. performed TEM experiment. The manuscript was mainly written by B.M. and Z.Y., and revised by T.S. The project was planned, directed and supervised by B.M. All authors discussed the results and commented on the manuscript.

## Additional information

**Competing financial interests:** The authors declare no competing financial interests.

