## [Peer Review File · Nature Communications]

Reviewers' Comments:

Reviewer #1 (Remarks to the Author)

I am happy with the changes the authors have made in response to my original review, and I can recommend the paper for publication in Nature Communications. I am glad that the authors spotted the error in their original PLQY calculations and have made the correction.

Reviewer #2 (Remarks to the Author)

This paper describes the synthesis and optical characterization of a 1D hybrid lead bromide material. The authors report blueish-white emission, which they ascribe to self-trapped exciton formation, in line with a number of recent papers on white light emission in 2D hybrid lead-bromide materials. The systems relatively high photoluminescence quantum yield is interesting. But I believe that significant revisions are required.

- 1) The authors claim that "1D structures with metal halide octahedra connected in a chain" have not been studied. This is not true. There are many reported 1D lead-bromide structures. Is the inorganic topology of this material novel? If not, prior examples must be cited.
- 2) The authors should not claim that their material "verifies the longstanding prediction that 1D systems are more favorable for exciton self-trapping" than the 2D or 3D cases. This is certainly not the first case of exciton self-trapping in a 1D material and this prediction has already been well studied across a number of 1D systems, including 1D metal-halides (ref. 25 and references therein).
- 3) The PLQE of 20% is for single crystals. This should be mentioned in the abstract. If this material were used in any application, the PLQE of powders (10-12%) or films would be more relevant.
- 4) Just because the barrier to self-trapping is small, does not necessarily mean that the photoluminescence quantum yield must be high. Until more information about this class of materials is acquired, it is misleading to state that the 1D nature of the structure leads to enhanced PLQE over 2D and 3D systems.
- 5) The statements relating non-radiative decay, temperature-dependent lifetime, and sample morphology are confusing, and are not supported well by the experiment (lines 169-178). Making arguments for the radiative and non-radiative decay rates for the free versus the self-trapped excitons (based on the data presented here) is impossible, as these emission processes cannot be separated.
- 6) The "configuration coordinate diagram" used to depict self-trapping in this system must cite prior literature. This same energy diagram has been used in much earlier papers that describe self-trapping.
- 7) If the structure is new, the CIF or the CIF deposition number should be given.
- 8) The lifetime of the material under continuous illumination (degradation rate) should be specified in the text.
- 9) The authors note that there are 3 self-trapped excited states, which justifies the use of the triexponential fit. However, no evidence is presented that suggests that there are only three different states, and recent work (ref 14.) in fact suggests a continuous distribution of excited states in 2D hybrid perovskites. The authors should explain why this fit was used.
- 10) The authors refer to "indirect self-trapped states". What does this mean? I have not seen this term in the literature.
- 11) There are a number of spelling and grammatical errors throughout the text.

Responses to Referees' Comments

Reviewer #1 (Remarks to the Author):

I am happy with the changes the authors have made in response to my original review, and I can recommend the paper for publication in Nature Communications. I am glad that the authors spotted the error in their original PLQY calculations and have made the correction.

We thank the reviewer for the recommendation.

Referee #2 (Remarks to the Author):

This paper describes the synthesis and optical characterization of a 1D hybrid lead bromide material. The authors report blueish-white emission, which they ascribe to self-trapped exciton formation, in line with a number of recent papers on white light emission in 2D hybrid lead-bromide materials. The systems relatively high photoluminescence quantum yield is interesting. But I believe that significant revisions are required.

1) The authors claim that "1D structures with metal halide octahedra connected in a chain" have not been studied. This is not true. There are many reported 1D lead-bromide structures. Is the inorganic topology of this material novel? If not, prior examples must be cited.

We appreciate the referee's comments. What we meant is that studies of 1D structures have been relatively rare as compared to 2D and 3D structures. We have revised the text and added couples of references (references 9-13) that reported 1D halid perovskites with metal halide octahedrons connected in a chain through corner sharing or face sharing. The 1D lead bromide perovskites presented in this work contain edge sharing lead bromide octahedrons, which are novel and discovered for the first time.

2) The authors should not claim that their material "verifies the longstanding prediction that 1D systems are more favorable for exciton self-trapping" than the 2D or 3D cases. This is certainly not the first case of exciton self-trapping in a 1D material and this prediction has already been well studied across a number of 1D systems, including 1D metal-halides (ref. 25 and references therein).

We agree with the referee that the prediction has already been studied in a few 1D systems. We have revised the main text in the abstract by stating "This work verifies once again that 1D systems are favorable for exciton self-trapping to produce highly efficient below-gap broadband luminescence".

3) The PLQE of 20% is for single crystals. This should be mentioned in the abstract. If this material were used in any application, the PLQE of powders (10-12%) or films would be more relevant.

We really appreciate the referee's recommendation. The abstract has been revised accordingly to show that the PLQE of ~ 20 % is achieved for the bulk single crystals.

4) Just because the barrier to self-trapping is small, does not necessarily mean that the photoluminescence quantum yield must be high. Until more information about this class of materials is acquired, it is misleading to state that the 1D nature of the structure leads to enhanced PLQE over 2D and 3D systems.

We agree with the referee that easy access to self-trapping excited states does not necessarily result in high PLQEs. We just reported the results we have observed, which showed higher PLQEs than 2D and 3D systems developed to date. We don't think we claimed that the 1D nature of the structure leads to enhanced PLQE over 2D and 3D systems in this manuscript. Instead, we just stated that 1D systems are favorable for exciton self-trapping to produce highly efficient below-gap broadband luminescence.

5) The statements relating non-radiative decay, temperature-dependent lifetime, and sample morphology are confusing, and are not supported well by the experiment (lines 169-178). Making arguments for the radiative and non-radiative decay rates for the free versus the self-trapped excitons (based on the data presented here) is impossible, as these emission processes cannot be separated.

We appreciate the referee's comments. To avoid the confusion, we have removed the discussions in lines 169-178. We are currently investigating the excited state dynamics and kinetics using ultrafast spectroscopies, which will help to give a better picture on the radiative and non-radiative decay processes.

6) The "configuration coordinate diagram" used to depict self-trapping in this system must cite prior literature. This same energy diagram has been used in much earlier papers that describe self-trapping.

Thanks for the referee's suggestion. We have cited the previous papers (references 22-24).

7) If the structure is new, the CIF or the CIF deposition number should be given.

CIF file has been uploaded with the revised manuscript.

8) The lifetime of the material under continuous illumination (degradation rate) should be specified in the text.

We believe that we have stated the photostability specifically in the text and in Supplementary Information Fig. 9. “These materials also show moderate photostability in air with slow decrease of emission intensity under continuous high power Hg lamp irradiation.” These materials are actually quite stable in the glovebox under ambient light. We are in process studying the degradation mechanism under strong UV irradiation in air.

9) The authors note that there are 3 self-trapped excited states, which justifies the use of the triexponential fit. However, no evidence is presented that suggests that there are only three different states, and recent work (ref 14.) in fact suggests a continuous distribution of excited states in 2D hybrid perovskites. The authors should explain why this fit was used.

We agree with the referee that there are multiple excited states (and possibly continuously distributed). Identifying the characteristics of these multiple excited states requires further investigations, for instance, ultrafast PL spectroscopy and transient absorption spectroscopy. In the present work, we used the triexponential fit for lifetime measurement that provided nice fitting and was consistent with the results reported previously in other systems. We don't think that we stated that there are only three self-trapped excited states in the manuscript, although the work cited had indicated three self trapped excited states (references 24, 25). The configuration coordinate diagram in Figure 3d is just a schematic diagram, which is not indicating only three self-trapped states get involved in the photophysical processes.

10) The authors refer to “indirect self-trapped states”. What does this mean? I have not seen this term in the literature.

Thanks for pointing this out. We have removed the term of “indirect” from the manuscript.

11) There are a number of spelling and grammatical errors throughout the text.

We have tried to correct the typos and grammatical errors.

Reviewers' Comments:

Reviewer #2 (Remarks to the Author)

I am mostly satisfied with the changes the authors have made in the revised manuscript and recommend publication after the following minor revisions.

The crystal's PLQE is comparatively high and this is interesting, but although it is a little lower, the powder PLQE is more important for any technology. So I recommend that both the single crystal PLQE and powder PLQE should be given in the abstract.

The section on the power-dependent photoluminescence study needs further clarification. The authors should report the energy delivered in $\mu\text{J}\cdot\text{pulse}\cdot\text{cm}^{-2}$ or $\text{photons}\cdot\text{pulse}\cdot\text{cm}^{-2}$, in addition to $\text{W}\cdot\text{cm}^{-2}$, as they are using a pulsed source. Based off the experimental section (1Hz rep rate, 5 ns pulse width), it is unclear if this number corresponds to the average power or the peak power of the incident light. If it is the average power, then the authors are pumping with an extremely high fluence ($>500 \text{ J}\cdot\text{cm}^{-2}$). If this number is the peak power, then (making assumptions about beam and pulse profile), the energy/pulse is actually small ($\mu\text{J}\cdot\text{pulse}\cdot\text{cm}^{-2}$). To prevent confusion, the authors should provide power dependence measurements of $\text{J}\cdot\text{pulse}^{-1}\cdot\text{cm}^{-2}$, and re-check their experimental details.

There are still a number of typos in the text.

Responses to Referees' Comments

Reviewer #2 (Remarks to the Author):

I am mostly satisfied with the changes the authors have made in the revised manuscript and recommend publication after the following minor revisions.

Response: We appreciate the reviewer's recommendation.

The crystal's PLQE is comparatively high and this is interesting, but although it is a little lower, the powder PLQE is more important for any technology. So I recommend that both the single crystal PLQE and powder PLQE should be given in the abstract.

Response: We have included the PLQEs for both bulk single crystals and microsize crystals in the abstract. In the revision, we have added "and ~ 12 % for the microscale crystals" behind "~ 20 % for the bulk single crystals".

The section on the power-dependent photoluminescence study needs further clarification. The authors should report the energy delivered in $\mu\text{J}\cdot\text{pulse}\cdot\text{cm}^{-2}$ or $\text{photons}\cdot\text{pulse}\cdot\text{cm}^{-2}$, in addition to $\text{W}\cdot\text{cm}^{-2}$, as they are using a pulsed source. Based off the experimental section (1Hz rep rate, 5 ns pulse width), it is unclear if this number corresponds to the average power or the peak power of the incident light. If it is the average power, then the authors are pumping with an extremely high fluence ($>500 \text{ J}\cdot\text{cm}^{-2}$). If this number is the peak power, then (making assumptions about beam and pulse profile), the energy/pulse is actually small ($\mu\text{J}\cdot\text{pulse}\cdot\text{cm}^{-2}$). To prevent confusion, the authors should provide power dependence measurements of $\text{J}\cdot\text{pulse}\cdot\text{cm}^{-2}$, and re-check their experimental details.

Response: To clear up the confusion, we have added "Peak to the label of the X axis of Figure 3a to read "Peak Power Density (W/cm^2)". We have also added a sentence "Peak power density (W/cm^2) was calculated by dividing the observed pulse intensity (in Joules) by the pulse width ($5\times 10^{-9} \text{ s}$) and the beam area in cm^2 ($\text{area} = \pi r^2 = \pi(0.25 \text{ cm})^2 = 0.20 \text{ cm}^2$). For example, $500 \text{ W}/\text{cm}^2$ peak power density was derived from an observed 490 nJ pulse intensity." after "100 collections were averaged at each power level with every laser pulse monitored to determine the average intensity. PL intensity was determined at the maximum of the PL emission curve." to clarify the experiment.

There are still a number of typos in the text.

Response: We have tried to correct the typos in the text, for instance, adding a "comma" before "respectively"; correcting the spellings, like "exited state" to "excited state", "emissio" to "emission", "reactoin" to "reaction", "degree" to "degrees", etc.